# Older Adults Who Maintained a Regular Physical Exercise Routine before the Pandemic Show Better Immune Response to Vaccination for COVID-19

**DOI:** 10.3390/ijerph20031939

**Published:** 2023-01-20

**Authors:** Brenda Rodrigues Silva, Fernanda Rodrigues Monteiro, Kizzy Cezário, Jônatas Bussador do Amaral, Vitória Paixão, Ewin Barbosa Almeida, Carlos André Freitas dos Santos, Gislene Rocha Amirato, Danielle Bruna Leal Oliveira, Edison Luiz Durigon, Andressa Simões Aguiar, Rodolfo P. Vieira, Juliana de Melo Batista dos Santos, Guilherme Eustáquio Furtado, Carolina Nunes França, Marina Tiemi Shio, André Luis Lacerda Bachi

**Affiliations:** 1Post-Graduation Program in Health Sciences, Santo Amaro University (UNISA), São Paulo 04829-300, Brazil; 2ENT Research Lab., Department of Otorhinolaryngology—Head and Neck Surgery, Federal University of Sao Paulo (UNIFESP), São Paulo 04021-001, Brazil; 3Discipline of Geriatrics and Gerontology, Department of Medicine, Paulista School of Medicine, Federal University of Sao Paulo (UNIFESP), São Paulo 04020-050, Brazil; 4Postgraduate Program in Translational Medicine, Department of Medicine, Paulista School of Medicine, Federal University of São Paulo (UNIFESP), São Paulo 04023-062, Brazil; 5Mane Garrincha Sports Education Center, Sports Department of the Municipality of Sao Paulo (SEME), São Paulo 04039-034, Brazil; 6Hospital Israelita Albert Einstein, São Paulo 05652-900, Brazil; 7Laboratory of Clinical and Molecular Virology, Department of Microbiology, Institute of Biomedical Science, University of São Paulo, São Paulo 05508-060, Brazil; 8Scientific Platform Pasteur, University of São Paulo, São Paulo 05508-060, Brazil; 9Infection Control Service, São Luiz Gonzaga Hospital of Santa Casa de Misericordia of São Paulo, São Paulo 02276-140, Brazil; 10Post-graduate Program in Human Movement and Rehabilitation and in Pharmaceutical Sciences, Universidade Evangélica de Goiás (Unievangelica), Av Universitária km 3,5, Anápolis-Go 75083-515, Brazil; 11Department of Physical Therapy, School of Medicine, University of São Paulo, São Paulo 05360-000, Brazil; 12Polytechnic Institute of Coimbra, Applied Research Institute, Rua da Misericórdia, Lagar dos Cortiços—S. Martinho do Bispo, 3045-093 Coimbra, Portugal

**Keywords:** COVID-19, vaccine, immunosenescence, antibody, lymphocyte, active lifestyle

## Abstract

Background: In this study, we aimed to investigate the specific-antibody response to the COVID-19 vaccination and the immunophenotyping of T cells in older adults who were engaged or not in an exercise training program before the pandemic. Methods: Ninety-three aged individuals (aged between 60 and 85 years) were separated into 3 groups: practitioners of physical exercise vaccinated with CoronaVac (PE-Co, *n* = 46), or vaccinated with ChadOx-1 (PE-Ch, *n* = 23), and non-practitioners vaccinated with ChadOx-1 (NPE-Ch, *n* = 24). Blood samples were collected before (pre) and 30 days after vaccination with the second vaccine dose. Results. Higher IgG levels and immunogenicity were found in the PE-Ch and NPE-Ch groups, whereas increased IgA levels were found only in the PE-Ch group post-vaccination. The PE-Co group showed a positive correlation between the IgA and IgG values, and lower IgG levels post-vaccination were associated with age. Significant alterations in the percentage of naive (CD28+CD57-), double-positive (CD28+CD57+), and senescent (CD28-CD57+) CD4+ T and CD8+ T cells were found post-vaccination, particularly in the PE-Ch group. Conclusions: The volunteers vaccinated with the ChadOx-1 presented not only a better antibody response but also a significant modulation in the percentage of T cell profiles, mainly in the previously exercised group.

## 1. Introduction

Since March 11, 2020, the World Health Organization (WHO) officially declared that the world is facing a pandemic due to the spread of the SARS-CoV-2 virus, which causes coronavirus disease-19, called COVID-19 [1]. Several different symptoms of COVID-19 were and are still being investigated [2,3,4]. However, it is known that the severity of COVID-19 can be closely associated not only with some clinical conditions of the individual infected by the SARS-CoV-2 virus but also with an exacerbated immune/inflammatory reaction to this infection, which induces a systemic hyperinflammatory response known as the “Cytokine Storm” [5]. Based on the fact that the inflammatory response can be involved in the severity of SARS-CoV-2 infection, epidemiological data highlight that older adults are among the populations most affected by COVID-19, in part due to the presence of a phenomenon named “inflammaging”, which is characterized as a chronic, sterile, low-grade systemic inflammatory profile and that can favor this population to present the highest rates of severe disease and death from SARS-CoV-2 infection [6].

In agreement with the literature, the occurrence of another phenomenon named immunosenescence, which is characterized by the gradual and significant reduction in immune responses associated with aging, is a pivotal factor that can also lead this population to present high fatality rates related to COVID-19 [7,8]. Among some aspects, this phenomenon is associated not only with a significant decline in the quantity and quality (neutralizing capacity) of antibodies produced in response to vaccination but also with a reduction in naive T cells (CD28+CD57-) in opposition to an increase in senescent T cells (CD28+CD57+), both for CD4 and CD8 T cells [6,7,8,9].

Based on the fact that the aging process profoundly impacts immune responses, nutrition and, especially, the regular practice of exercise training, a well-known non-pharmacological intervention, have received great scientific attention as they are able to minimize the development and progression of immunosenescence [10,11]. In association or not with other therapies, exercise training is commonly recommended to older adults due to its capacity to benefit several aspects, which can include the improvement of the immune response to vaccination, as previously reported by our group [12,13], as well as acting as an adjuvant factor in combating infectious diseases, particularly leading to a reduced risk for viral respiratory infections, including SARS-CoV-2 [14,15,16].

Despite being reported that exercise training could be an important tool to improve/maintain the immune response in the COVID-19 era [17], the impact of social isolation imposed by the pandemic on the immune response, which includes both humoral and cellular responses, of older adults to COVID-19 vaccination is not fully understood. The most recent findings revealed that exercise reduces the negative effects of isolation, such as stress, anxiety, and sedentarism, all of which reduce immunity and increase the risk of noncommunicable disease [18]. It is strongly desirable to increase host immunity and reduce the harmful consequences of isolation through exercise in the COVID era and beyond [17,19]. During the current pandemic, which is a demanding environment in terms of nutrition, psychology, and social interaction due to the presence of a virulent viral organism, it is recommended that exercise training should be performed at moderate intensities and volumes [20]. In fact, it has been demonstrated that whilst moderate exercise can improve health status, an inadequate performance of intense exercise can favor the occurrence of numerous pathologies, including COVID-19, due to the malfunction of the immune system [21].

Although engagement in physical exercises programs, such as combined-exercise training, can improve the immune response of older adults to vaccination [22], the restrictions imposed by the pandemic, especially for this population, obligated them to alter their routine, and the maintenance of a regular practice of exercise training was impacted [23], as also evidenced by Brancaccio and collaborators in a study involving young adults [24]. Because Influenza virus vaccination (IVV) was the most commonly used vaccination in the acute-exercise literature, this relationship was limited to IVV in older adults engaged in acute short-term and quasi-experimental studies [22]. However, the results of a very recent study showed that 90 min of moderate aerobic exercise consistently increased serum antibody to both Influenza virus and COVID-19 vaccines four weeks post-immunization [20]. These findings are consistent with previous research [25,26], including some conducted by our research group [27,28], and indicate that regular exercise training is able to minimize the risk of respiratory infections, such as those induced by the Influenza virus and SARS-CoV-2 [16].

It is consensus that vaccination against SARS-CoV-2 is, until now, the best option not only to protect the global population, especially aged people, but also to put the pandemic to an end [29,30,31]. In relation to the vaccines for COVID-19 used in Brazil, the first vaccine approved was CoronaVac (BBIBP-CorV, Sinovac Biotech), a vaccine that contains an inactivated whole virion [32], and is currently produced by the Instituto Butantã. The second vaccine approved was the ChAdOx1 nCoV-19 (AZD1222, Oxford-AstraZeneca) vaccine, a vaccine that is composed of a replication-deficient adenoviral vectored vaccine that encodes the SARS-CoV-2 spike protein [32], and is currently produced by the Fundação Oswaldo Cruz (Fiocruz). Beyond these vaccines, others such as the Pfizer BNT162b2 SARS-CoV-2 mRNA vaccine and Ad26.COV2.S were also approved and made available for the Brazilian population.

Even though the use of these vaccines has promoted the relaxation of pandemic-related restrictions, in this study, we aimed to investigate the impact of one year of social isolation imposed by the pandemic on the antibody response to COVID-19 vaccination. In addition, the percentages of naive and senescent T cells in groups of older adults who were engaged, or not, in a program of exercise training before the pandemic were investigated, in order to better understand the effects of the abrupt interruption of the regular practice of exercise training on the immune response of older adults.

## 2. Materials and Methods

### 2.1. Study Design

This is a multifactorial, interventional, and prospective pre-post 3-arm controlled trial study, involving older adults (aged between 60 and 85 years) who were vaccinated for COVID-19 and were engaged or not in a program of combined exercise training, for at least 18 months, before the social isolation imposed by COVID-19. Volunteers of both genders were recruited from the Geriatrics and Gerontology Discipline of the Federal University of São Paulo (UNIFESP) or from the “Hospital Geriátrico e de Convalescentes Dom Pedro II”, belonging to the Health Department of the São Paulo State, Brazil. Data and blood samples were collected between January and February 2021 (before the administration of the first dose of the vaccine for COVID-19) and between March and April 2021 (30 days after the administration of the second dose of the CoronaVac vaccine) or between May and June 2021 (30 days after the administration of the second dose of the ChadOx-1 vaccine). It is noteworthy to mention that this difference in terms of sample collection was associated with the vaccination schedule for CoronaVac and ChadOx-1 as, for CoronaVac, the second dose was administered around 28 days after the first dose was received, and for ChadOx-1, the second dose was administered around 120 days after the first dose was received. A scheme of design is presented in Figure 1. The Transparent Reporting of Evaluations with Nonrandomized Designs (TREND) was followed to guide this purpose [33].

### 2.2. Ethical Statement

All participants signed the informed consent form previously approved by the Ethics Committee of the University of Santo Amaro (approval number 4,350,476 and 4,951,537) and the University of São Paulo (USP, under number 36011220.0000.0081). The study respects the Brazilian Resolution (196/96) on ethics in research with humans [34], follows the guidelines for ethics in scientific experiments in exercise science research [35], and also complies with the Helsinki Declaration guidelines for research with humans [36].

### 2.3. Participants’ Selection Criteria

The following criteria for inclusion have been applied: (1) take part in the study spontaneously; (2) be 60 years of age or older, with both sexes permitted to participate; (3) have the autonomy to move from their residence to the exercise program’s center; (4) obtain medical clearance to engage in exercise sessions in combination with vaccination. Exclusion criteria were also established: (1) have medication or disease control on a daily basis that could interfere with the practice of regular exercise; (2) have received a diagnosis of any type of severe mental and physical illness or presented acute or chronic infections, neoplasms, or liver and renal diseases; (3) have submitted to pharmacological therapy with anti-inflammatory drugs or with convalescent plasma during any stage of the study.

It is noteworthy to mention that none of the volunteers reported being previously infected by SARS-CoV-2 until the study began or presented any symptom associated with SARS-CoV-2 infection during the development of the present study.

In addition, it is also important to cite that, during the study period, all volunteers were oriented to maintain their daily activity routines.

#### Sample Size Calculation and Experimental Groups

Although this experimental design was planned as a naturalistic and exploratory trial study, the sample calculation was performed to establish a minimum number of participants per group. Using the G*Power software program [37], the sample size and statistical power were estimated, based on the Student’s T-test, effect size (0.30) at α-level (0.05), statistical power of 0.95, and three groups with two (pre-pos) measures. Considering a margin of 30% losses or refusal [38], a minimum of 22 individuals per group were putative. As shown in the flowchart (Figure 2), a total of 102 older adult individuals from the community were initially invited to take part in the study. Together with the medical and academic staff, and guided by participant selection criteria previously determined, our research team recruited 96 volunteers, which were separated into three subgroups according to the type of vaccine received, and their engagement in the physical exercise program before the pandemic. The following groups were generated from the final sample analysis: (i) two groups, one composed of older adults previously engaged in a program of physical exercise (PE) who were vaccinated with CoronaVac (PE-Co, *n* = 46), and another group composed of older adults who were vaccinated with ChadOx-1 (PE-Ch, *n* = 23); (ii) one group of older adult individuals non-engaged in a program of physical exercise (NPE) who were vaccinated with the ChadOx-1 vaccine (NPE-Ch, *n* = 24).

### 2.4. Exercise Intervention Protocol

Specific details about the exercise training program performed by the older adult volunteers who composed the exercise training program can be evidenced in our previously reports [27,39]. Briefly, this population performed the following program that consisted of a combination of continuous aerobic and resistance exercises. Each session had a duration of 60–75 min, performed three times a week on alternate days, in moderate-intensity monitoring by assessing by means of a heart rate monitor (Polar brand, model FT1, Polar-Finland) and by the Borg Scale of Perceived Exertion. All of them performed the program for, at least, 48 weeks (12 months), and were supervised by the same physical education professional.

### 2.5. Vaccination for COVID-19

In agreement with the vaccination schedule initially proposed to the older adult population by the National Health Surveillance Agency (ANVISA) of Brazil, in January–February, 2021, this population could be exclusively immunized with CoronaVac or ChadOx-1. Based on it, the volunteers enrolled in this study were immunized with the vaccines for COVID-19 available by the Unified Health System (SUS), Brazil. It is noteworthy to clarify that, at those times, as formerly described, all volunteers were submitted to two doses of the same vaccine, in which for CoronaVac, the interval between the first and second doses was around 28 days, and for ChadOx-1, it was around 120 days.

### 2.6. Blood Sample Collection

Blood samples were collected on two different occasions: before (pre) and 30 days after (post) administration of the second dose of the vaccine for COVID-19, both for CoronaVac and for ChadOx-1. One blood sample was collected in a tube without any anticoagulant compound to obtain sera aliquots, which were used to determine the specific IgA and IgG antibodies levels for SARS-CoV-2 antigens. Two other blood samples were collected in tubes containing anticoagulant EDTA to obtain the peripheral blood mononuclear cells (PBMCs), which were used in T cells immunophenotyping analysis, as described in the following.

### 2.7. Determination of Specific Antibodies (IgA and IgG) for the SARS-CoV-2 Antigens

Serum aliquots (minimum 500 μL) were obtained after blood clotting in the collection tube itself and centrifuged at 2500 rpm for 10 min at 4 °C, being subsequently frozen at −80 °C. Later, the concentration of SARS-CoV-2-specific IgA and IgG was determined through the ELISA technique on serum samples obtained before and after vaccination. To perform this test, we followed the procedure previously described by our group [40], in which the concentration of SARS-CoV-2 N, M, and S antigens (0.12 µg/mL) was used to sensitize the reaction plates. After the blockage and washing steps, the samples were diluted 1:2000 for IgA and 1:5000 for IgG in 0.1% PBS-tween buffer containing 1 M NaCl, added to the plate, and incubated for at least 2 h at 37 °C. After this incubation and washing step, anti-IgA (1:2000) and anti-IgG (1:10,000) conjugates for humans were added to the plate and maintained for 1 h at 37 °C. After this time, a new washing step was performed, and the reaction was evaluated by adding a solution of citrate buffer (pH 4.3) plus OPD and H_2_O_2_ to each well of the plate. Then, the reaction was stopped with the addition of 2N H_2_SO_4_ solution. The reaction reading was performed at 492 nm in a microplate reader (Labsystem Mulitskan MS, Artisan Technology Group: Champaign, IL, USA).

### 2.8. Immunophenotyping of T Cells

Briefly, after sampling collection, initially, the blood was mixed 1:1 with phosphate-buffered saline (PBS 1×, pH = 7.4) and the peripheral blood mononuclear cells (PMBCs) were obtained after centrifugation of the Falcon^®^ tube containing blood previously diluted and Ficoll-Hypaque (GE Healthcare Bio-Sciences AB, Uppsala, Sweden). Then, 1 × 106 cells were mixed with 1 mL of freezing medium (90% fetal bovine serum + 10% DMSO) and storage in liquid nitrogen. The T cell immunophenotyping was carried out after the thawing of PBMC. Afterward, PBMC was transferred to 1.5 mL Eppendorf tubes with MACS Buffer (PBS (1×, pH = 7.4) containing albumin of fetal bovine serum (0.5%) and EDTA (2 mM)), with a final volume of 250 μL. Following this step, the cells were incubated with the monoclonal antibody mix to perform the immunophenotyping of CD4+ T cells and CD8+ T cells expressing or not CD28 and CD57 molecules, through the immunostaining using the monoclonal antibodies for Flow Cytometry assays: anti-CD4 FITC, anti-CD57 PE, and anti-CD28 PercP. PBMC was incubated at 4 °C for 30 min protected from light, and, after this incubation, the cells were submitted to two successive washes with MACS buffer. Finally, the cells were submitted to the flow cytometer FACSCalibur™ (Becton Dickinson Immunodiagnostic Systems, San Jose, CA, USA). Data generated were analyzed in the Cell Quest Pro program (Becton Dickinson Immunodiagnostic Systems, San Jose, CA, USA). The definition of the gates was based on the analysis of the FSC-H and SSC-H (Forward Scatter and Side Scatter, respectively) and the analysis strategies are shown in Figure 3.

## 3. Statistical Analysis

Initially, the data obtained were compared with the Gauss curve and the normality for each was determined by the Shapiro–Wilk test. In addition, the homogeneity of variance was evaluated by Levene’s test. As all variables showed a non-parametric behavior, they were represented by median and interquartile ranges. In addition, intragroup comparisons were performed using the Wilcoxon test, whereas the intergroup evaluations were performed using the Kruskal–Wallis test with the Müller–Dunn post hoc test. The Spearman’s rank correlation coefficient test was used to assess the existence of correlation among the variables studied. Lastly, the multivariate regression analysis adjusted for age was also performed. The α risk considered in this study was set at 5% (*p* < 0.05).

## 4. Results

### 4.1. Sample Characterization

Table 1 presents the anthropometric data of the volunteers from the PE and NPE groups submitted to the vaccination with CoronaVac or ChadOx-1, as well as the number of women and men that composed these groups. According to our results, only the number of women and men in the subgroups showed a significant difference.

### 4.2. Specific Antibodies of (IgA and IgG) for the SARS-CoV-2 Antigens

Figure 4 shows the results regarding serum levels of specific IgG (Figure 4A) and IgA (Figure 4B) for SARS-CoV-2 antigens obtained pre- and post-vaccination. It was found that serum levels of IgG and IgA significantly increased post-vaccination in the volunteers of the PE group vaccinated with ChadOx-1 (PE-Ch) as compared to pre-vaccination values (*p* = 0.0002 and *p* = 0.0001, respectively), whilst the volunteers of the PE group vaccinated with CoronaVac (PE-Co) did not show differences in these antibodies post-vaccination in comparison to the pre-vaccination values (IgG, *p* = 0.4764 and IgA, *p* > 0.9999). Regarding the results obtained in the NPE group, which was exclusively submitted to the ChadOx-1 vaccine (NPE-Ch), a significant increase only in serum levels of IgG post-vaccination (*p* = 0.0012) was observed, whereas the IgA levels were unchanged (*p* = 0.0645).

As the number of men and women enrolled in the groups vaccinated with ChadOx-1 showed significant differences, we performed an evaluation of the specific antibody response (IgG and IgA) for SARS-CoV-2 antigens separating the volunteers by gender. As presented in Appendix A (SF1), the intragroup analysis showed higher specific IgG levels both in the subgroup of older men who composed the NPE group (SF1-A, *p* = 0.02) and older women who composed the PE group (SF1-A, *p* = 0.01) post-vaccination than the values pre-vaccination. In relation to specific IgA levels SF1-B), both subgroups of older men and older women from the PE group showed significantly increased levels of this antibody post-vaccination as compared to the value’s pre-vaccination (*p* = 0.03 and *p* = 0.006, respectively). In the intergroup analysis, no differences were found between the subgroups of men and women who composed NPE and PE groups.

Although no participants reported any symptoms or previous infection by SARS-CoV-2, some volunteers in both volunteer groups, but mainly in the volunteer group vaccinated with CoronaVac, presented antibodies for SARS-CoV-2 antigens during the pre-vaccination occasion time. Based on the fact that these data could interfere with our results, we performed an evaluation concerning the immunogenicity related to the serum response of specific IgG (Figure 5A) and IgA (Figure 5B) for SARS-CoV-2 antigens in the volunteer groups participating in this study

In this respect, the immunogenicity for IgG (Figure 5A) and IgA (Figure 5B) in the PE group vaccinated with CoronaVac (PE-Co group) was as follows: 39.1% and 19.6% responded to vaccination as they presented an increase in their levels post-vaccination, whereas 39.1% and 71.7% did not respond, and also 21.8% and 8.7% had a drop in total serum concentrations of these antibodies post-vaccination as compared to pre-vaccination values, respectively. Concerning the results obtained for serum levels of IgG (Figure 5A) and IgA (Figure 5B) in the PE group vaccinated with ChadOx-1 (PE-Ch group), it was found that 56.5% and 60.9% responded to vaccination, whilst 43.5% and 39.1% did not respond to the vaccination, respectively. However, in the NPE group vaccinated with ChadOx-1 (NPE-Ch group), the results obtained for serum levels of IgG (Figure 5A) and IgA (Figure 5B) were as follows: 66.7% and 33.3% responded to vaccination, whereas 20.8% and 58.3% did not respond, and also 12.5% and 8.4% had a drop in the concentrations of these antibodies post-vaccination as compared to the pre-vaccination values, respectively.

In order to verify the systemic bioavailability of IgG and IgA both before (pre) and post-administration of the vaccines CoronaVac (PE group, Figure 6A,B, respectively) or ChadOx-1 (PE group, Figure 6C,D; NPE group, Figure 6D,E, respectively) in the volunteers participating in the study, we performed Spearman’s rank correlation coefficient test. As presented in Figure 6, only the PE group vaccinated with CoronaVac (Figure 6B) showed a significant positive correlation between serum levels of IgG and IgA post-vaccination time-point. No other significant result was found.

### 4.3. Immunophenotyping of T Cells

Beyond the antibody responses to COVID-19 vaccination, we also assessed the percentage of CD4 T cells and CD8 T cells not only naive (CD28+CD57-) but also double-positive (CD28+CD57+) and senescent (CD28-CD57+), before and 30 days after administration of the second dose of CoronaVac and ChadOx-1 vaccines. However, it is important to point out that this analysis was only performed in the PE group because, unfortunately, we did not have access to blood samples collected with an anticoagulant from volunteers in the NPE group.

As shown in Figure 7, a lower percentage of naive cells (CD28+CD57-), both for CD4+ and CD8+ T cells, was observed in the PE group vaccinated with ChadOx-1 (Figure 7A, *p* = 0001; Figure 7B, *p* = 0.0006, respectively), whereas, in the PE group vaccinated with CoronaVac, a lower percentage of naive CD4+ T cells was found (Figure 7A, *p* = 0.0231) and no differences in the percentage of naive CD8+ T cells (Figure 7B, *p* = 0.0231) post-vaccination were found as compared to the pre-vaccination values. Concerning the percentages of CD4+ T and CD8+ T cells double-positive for CD28 and CD57, the PE group vaccinated with ChadOx-1 showed a significant increase post-vaccination as compared to pre-vaccination values (Figure 7C, *p* = 0.0045; Figure 7D, *p* = 0.0172, respectively), whilst the values observed in the PE group vaccinated with CoronaVac were unchanged (CD4+ T cells—Figure 7C, *p* = 0.2003; for CD8+ T cells—Figure 7D, *p* = 0.7475). Interestingly, in the intragroup analysis related to the percentage of senescent cells (CD28+CD57-), both for CD4+ T and CD8+ T cells, no differences were found in these percentages both in the PE group vaccinated with ChadOx-1 (Figure 7E, *p* = 0.7919; Figure 5F, *p* = 0.0956) and in the PE group vaccinated with CoronaVac (CD4+ T cells—Figure 7E, *p* = 0.3585; CD8+ T cells—Figure 7F, *p* = 0.0936) in the comparison of the values observed pre- and post-vaccination. However, in the intergroup analysis, a significant increase in the percentages of senescent CD8+ T cells post-vaccination was found in the PE group vaccinated with CoronaVac as compared to the values observed in the PE group vaccinated with ChadOx-1 (Figure 7F, *p* = 0.007).

As age can impact the immune response, we additionally performed a multivariate regression analysis adjusted for age, and it is possible to observe that age presented a significant effect on the serum levels of specific IgG for SARS-CoV-2 antigens post-vaccination for only the PE group vaccinated with CoronaVac (β = −0.01536; 95% CI −0.02970 to −0.001029; *p* = 0.0363; R^2^ = 0.2296).

## 5. Discussion

This study aimed to examine the specific-antibody response to the COVID-19 vaccination and the percentage of both naïve and senescent T cells in older adults who maintained or not a regular practice of exercises before the pandemic period. Generally, the findings obtained from this study showed that: (1) the ChadOx-1 vaccine was more effective in eliciting an immune response, particularly in terms of immunogenicity, than the CoronaVac vaccine; (2) the group of older adults who were engaged in a combined-exercise training program before the pandemic presented a better vaccination response, mainly in terms of specific IgG and IgA for SARS-CoV-2 antigens than the group composed of older adult non-practitioners of a physical exercise program.

It is broadly accepted that the immunosenescence process drives significant alterations not only in the number but also in the functionality of immune cells. For instance, this process can negatively impact both B cell responses, the cells responsible of producing antibodies against a specific antigen, leading to a reduced humoral response, as well as T cells, promoting a decline in the number of naive T cells and accumulation of senescent T cells in association with remarkable alterations in their functionality [41]. As these alterations are presented, the immune responses to vaccination in older adults will be impaired [28]. Corroborating these pieces of information, it was documented that the annual vaccination for the Influenza virus, the pathogen that causes flu, has demonstrated an immunogenicity around 40% to 50% in older adults, which leads this population to present a high rate of infections, especially by respiratory viruses, even in the vaccinated individuals [42].

Beyond the immunosenescence, at this point, it is paramount to highlight that, in agreement with the literature, both quarantine and social distancing are considered unpleasant experiences, which involve loss of freedom, loneliness, job uncertainty, separation from loved ones, and fear of illness due to several restrictions that abruptly upset the lifestyle [21,43,44]. Corroborating these pieces of information, it was demonstrated that the social isolation imposed by COVID-19 increased the sedentary lifestyle [45], both in men and women populations [24], which can lead to a weakening of the immune system to SARS-CoV-2 infection and also vaccination.

Despite aging being a natural process and that the occurrence of immunosenescence cannot be completely avoided, both the clinical practice and scientific literature, every day, highlight that the regular practice of exercise training can be considered a powerful tool to minimize the decline of the immune system associated with aging, which includes an improvement in the immune responses to vaccination [12,20].

In this respect, our group has reported that older adults who regularly practice a moderate intensity of a combined-exercise training program presented a better antibody response to vaccination against the Influenza virus, both systemically and in the mucosa of the upper airways, as compared with older adults who did not practice exercise training or presented a sedentary lifestyle [27,28]. Based on these data, the results obtained in the present study allow us putatively to suggest that the benefits promoted by the long-standing regular practice of exercise training on the immune responses of older adults, as far as we could evaluate, were maintained even after the interruption of its practice for a year due to the social isolation imposed by the pandemic of COVID-19. This suggestion is based on the observation that the PE group vaccinated with ChadOx-1 not only presented a more pronounced increase in the levels of specific IgG (*p* = 0.0002) than the values observed in the NPE group (*p* = 0.0012), but also that only the volunteers in this group showed a significant increase in IgA levels post-vaccination as compared to the pre-vaccination values.

In accordance with the literature, most vaccines against respiratory viruses, such as COVID-19, are administered, in general, intramuscularly in order to elicit a robust systemic immune response, which includes higher IgG levels, and not a protective immunity on the mucosa of the upper airways, mainly by secretory IgA, as can be developed after natural infection by the respiratory virus [46]. Although it is known that serum IgA levels present a minor impact on immune protection in the mucosa [47,48], it is not fully understood whether serum IgA levels could be correlated with an adequate IgA response in the mucosa [49], improving the immune response to infection by SARS-CoV-2, as it was reported that IgA antibody responses were associated with lower viral shedding [49]. Furthermore, it is noteworthy to mention that higher IgA levels were related to an increase in the efficacy of IVV [50]. Therefore, these pieces of information can support our suggestion that the PE group vaccinated with ChadOx-1 presented a better antibody response than those found in the NPE group, which was submitted to the same vaccination schedule.

Despite the PE group presenting a more prominent antibody response to the vaccination with ChadOx-1 than the NPE group, which could be putatively attributed to their previous lifestyle, in an interesting way, the immunogenicity analysis, particularly in terms of serum IgG levels, showed an opposite result between these groups. In fact, the immunogenicity found in the NPE group was higher (66.7%) than the values found in the PE group (56.5%), which can demonstrate that the interruption of the regular practice of exercise training for one year could negatively impact the immune response of some older adults to vaccination. By the way, our group recently published a study reporting that the social isolation imposed by the pandemic was able to significantly alter some metabolic parameters and worsen the functional physical capacity of an older women group previously exercised [51]. It is of utmost importance to cite that the immunogenicity found in the groups vaccinated with ChadOx-1, regardless of the previous practice or not of exercise training, was above that frequently observed for older adults, which is around 40–50%, as formerly mentioned [42].

Corroborating this last information, the immunogenicity found in the volunteers of the PE group vaccinated with CoronaVac was 39.1%, even though a study performed with aged individuals (≥60 years) in Chile had demonstrated that, after 4 weeks of administration of the second dose of CoronaVac, the immunogenicity found in this population was 70.37% [52]. Despite not being able to affirm, this lower immunogenicity found in the PE group vaccinated with CoronaVac could putatively be attributed to the fact that only in this group was a negative effect of age also found in the serum levels of specific IgG for SARS-CoV-2 antigens post-vaccination. Another point that deserves to be mentioned was that the same group presented a positive correlation between the serum levels of specific IgG and IgA for SARS-CoV-2 antigens post-vaccination, which indicates that, in some volunteers, the production of IgG and IgA occurred concomitantly, a fact that could have been impacted by the previous natural infection by SARS-CoV-2, as it was suggested that pre-existing antibodies can negatively interfere with the ability to respond to repeated antigenic stimulation [52]. It is important to highlight that in this research, this group had the majority of asymptomatic cases of prior SARS-VCoV-2 infection.

Beyond the specific antibody responses, the technologies applied to produce the different vaccines for COVID-19 can also elicit a cellular response. Beyond the specific antibody responses, the technologies applied to produce the different vaccines for COVID-19 can also elicit a cellular response. In fact, studies have highlighted that T cells responses showed a close correlation with clinical protection in older adults [53,54,55]. For instance, the presence of specific subsets of CD4+ and CD8+ T cells for internal proteins of Influenza virus showed an evident correlation with better outcomes for this infection [56,57,58]. In this respect, whilst the CoronaVac vaccine is composed of an inactivated whole virion [32], the ChadOx-1 vaccine is composed of a replication-deficient chimpanzee adenovirus vector that carries the gene of the spike structural surface protein of the SARS-CoV-2 virus [59]. It has been reported that these vaccines are able to induce not only humoral but also cellular responses [52,60,61], which corroborates our findings concerning T cell responses, both CD4+ and CD8+ T cells, to vaccination assessed here.

In an interesting way, both vaccines applied in the PE group were able to induce a significant reduction in the naive CD4+ T cells. In accordance with the literature, the presence of cellular response has a corollary action in the maintenance of a robust response against virus infection, not only related to a re-exposure but also to the presence of new variants of this virus. Moreover, the collaboration between T and B cells is also crucial for the development of persistent immunity to virus infection, including SARS-CoV-2 infection [26]. Of interest, it is pointed out that a weakened CD4+ T cell response will interfere in the development of a protective immune response to vaccination in older adults [58]. It is well-known that naive CD4+ T cells are involved in the induction of cellular responses for new antigens; thus, the significant reduction in these cells found here allows us to putatively suggest that the immunological challenge imposed by the vaccination for COVID-19 in the volunteers of the PE group was able to activate these cells, improving the conditions of B cells to produce antibodies due to their function of the helper immune cells [62,63].

Although it was demonstrated that the CoronaVac vaccine was able to induce an increase in the number of CD4+ T cells and CD8+ T cells, particularly associated with the effector memory profile, in older adults [27], in our study, the vaccination with this vaccine only impacted the percentage of naïve CD4+ T cells. In contrast, the ChadOx-1 vaccine demonstrated the capacity to impact not only CD4+ T cells but also CD8+ T cells, including different profiles assessed in this study (naive and double-positive for CD28+ and Cd57+).

In terms of naive CD8+ T cells, as previously mentioned for CD4+ T cells, the reduction in these cells can reinforce our suggestion that a cellular response was elicited by the vaccination with ChadOx-1 and can also indicate that an anti-viral response associated with CD8+ T cells was properly generated [64], leading to an improvement of cellular immunity against SARS-CoV-2 infection. Despite not being able to affirm that the reduction in naive T cells, both of CD4+ and CD8+ T cells, was directly related to an increase in the number of T cells presenting a classical effector profile, our findings of an increase in the percentages of CD4+ T and CD8+ T cells expressing CD28+ and CD57+ can help us to suggest that an improvement of immune response was elicited by the ChadOx-1 vaccine, as it was reported that the increase in T cells double-positive for these markers, particularly in CD8+ T cells, was involved in the response to vaccination for hepatitis B. In addition, it was also reported that double-positive CD8+ T cells have the capacity to produce several cytokines, especially IL-10, a well-known anti-inflammatory cytokine; these cells were decreased in patients with rheumatoid arthritis, and an increase in the frequency of these cells in old age can benefit the immune responses [65]. In accordance with the description presented in a previous study [65], the markers CD28 and CD57 can be used to characterize four distinct profiles of T cells: CD28+CD57−, which indicates non-activated or early-activated cells; CD28+CD57+ is related to activated cells; CD28−CD57− can indicate activated or early-senescent cells; CD28−CD57+ is indicative of terminally differentiated-senescent-like cells.

According to the literature, aged individuals present a gradual accumulation of senescent immune cells, with a highlight on T cells, and the existence of these cells is involved with the development of an age-related pro-inflammatory systemic phenotype, named *inflammaging*, which refers to a chronic, systemic, low-grade inflammation associated with aging [66,67]. Moreover, there is a handful of pieces of evidence demonstrating that the long-term persistence of senescent cells can be implicated in systemic detrimental effects as the accumulation of these cells was associated with many chronic disease states and poor vaccination responses [68,69,70,71].

Considering that the presence of senescent cells can impair vaccination responses in older adults, we can putatively suggest that the response found in this study to the ChadOx-1 vaccine was better than that found after vaccination with CoronaVac as it was verified that the PE group vaccinated with CoronaVac presented higher percentages of senescent CD8+ T cells than the PE group vaccinated with ChadOx-1. In consequence, the higher percentage of senescent T cells in the PE group vaccinated with CoronaVac could be putatively involved in the poor vaccination response of this group as compared to the PE group vaccinated with ChadOx-1.

## 6. Limitations of the Study

The limitations of the study were: (1) The discrepant number of volunteers in the PE group vaccinated with CoronaVac (*n* = 52) and the other volunteer groups vaccinated with ChadOx-1 (23 in the PE group and 24 in the NPE group). In this respect, it is important to mention that this difference could be attributed to the fact that (1) the first vaccine for COVID-19 available in Brazil was CoronaVac, which was mainly intended for the older adult population, thus leading to the majority of the volunteers enrolled in this study to be immunized with this vaccine, and also that, (2) as formerly cited, the volunteers enrolled in the NPE group were exclusively recruited from the older adults who lived in the “Hospital Geriátrico e de Convalescentes Dom Pedro II”, an institution in which the aged people can live for a long period, similarly to an asylum, and, for them, were administered only the ChadOx-1 vaccine. (2) A group of older adults’ non-practitioners of exercise training vaccinated with CoronaVac could not only improve our understanding of the immunogenicity of this vaccine in the older adult population but also verify whether the long-standing regular practice of exercise training could favor the older adults to present a satisfactory immune response to the vaccination with CoronaVac. (3) The lack of information on whether the participants who presented specific levels of IgG and IgA for SARS-CoV-2 antigens before the COVID-19 vaccination were really infected by SARS-CoV-2 or other coronaviruses that cross-react with SARS-CoV-2 antigens. (4) The lack of comparison of T lymphocyte immunophenotyping between PE and NPE groups, which could improve the present study findings. (5) The discrepant number of women and men who composed the volunteer groups, although the data presented in the Appendix A can minimize this drawback. (6) The lack of assessment of the systemic levels of both pro- and anti-inflammatory cytokines, which could improve our understanding of the immunological status of our volunteers as, as formerly mentioned, inflammaging is a phenomenon that can impact vaccination responses.

## 7. Conclusions

The benefits obtained from a regular practice of a combined-exercise training program, even after one year of interruption, can improve the specific immune response to the vaccination for COVID-19. In addition, our findings demonstrate that older adults vaccinated with ChadOx-1 had better responses to both specific antibodies and T cells than those observed in the older adult group submitted to vaccination with CoronaVac.

## Figures and Tables

**Figure 1 ijerph-20-01939-f001:**
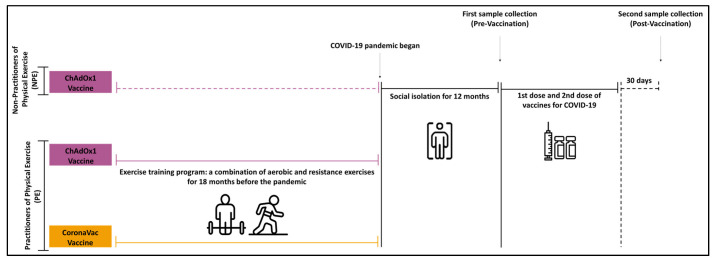
Representative scheme of study design.

**Figure 2 ijerph-20-01939-f002:**
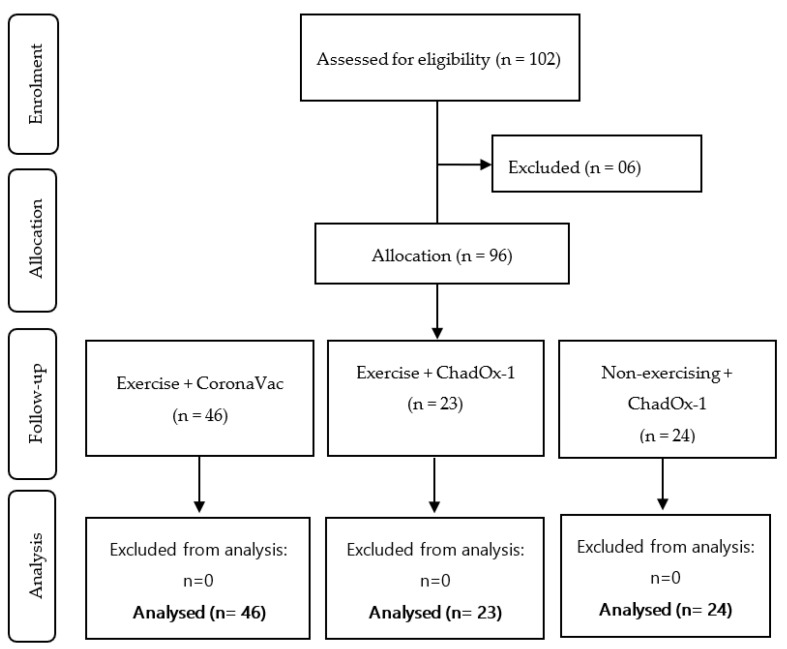
Flowchart of the study design.

**Figure 3 ijerph-20-01939-f003:**
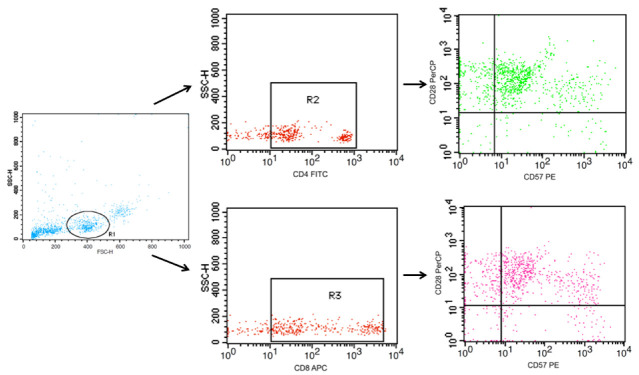
Representative dot plots of a flow cytometry panel used for the detection of both CD4+ T cells and CD8+ T cells associated with the profile naïve (CD28+CD57−), double-positive (CD28+CD57+), and senescent (CD28-CD57-). PBMCs were stained with Abs recognizing CD4, CD8, CD28, and CD57 and were analyzed by flow cytometry, with the gating as indicated.

**Figure 4 ijerph-20-01939-f004:**
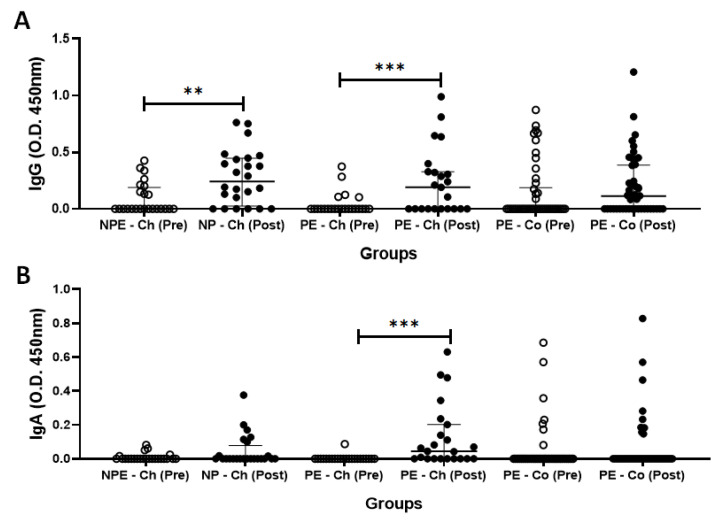
Total serum concentration (O.D. 450 nm) of (**A**) specific IgG and (**B**) specific IgA for SARS-CoV-2 antigens before (pre) and after 30 days (post) of administration of the second dose of ChadOx-1 and CoronaVac vaccine in the groups of older adults who regularly practiced (PE) or not (NPE) a physical exercise program before the pandemic period. Data are presented as median and interquartile range. ** *p* < 0.01; *** *p* < 0.001.

**Figure 5 ijerph-20-01939-f005:**
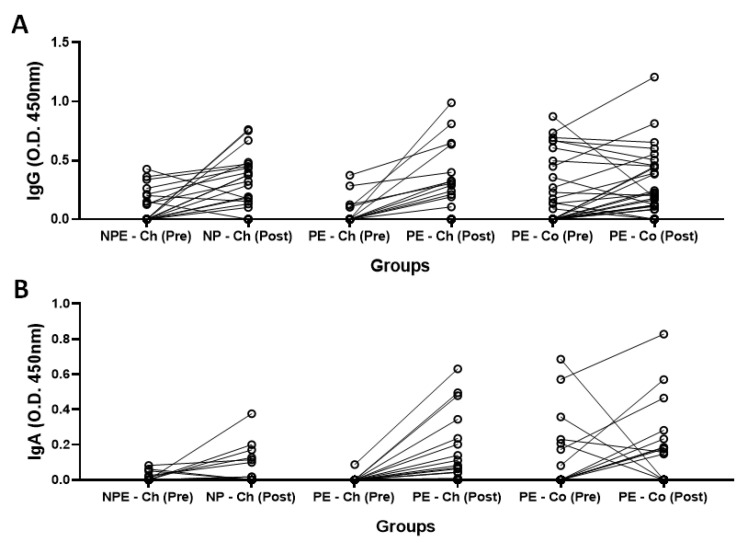
Representation of the antibody response of each volunteer, in terms of the serum concentration (O.D. 450 nm) of specific IgG (**A**) and IgA (**B**) for SARS-CoV-2 antigens before (pre) and after 30 days (post) of administration of the second dose of ChadOx-1 and CoronaVac vaccine in the groups of older adults who regularly practiced (PE) or not (NPE) a physical exercise program before the pandemic period.

**Figure 6 ijerph-20-01939-f006:**
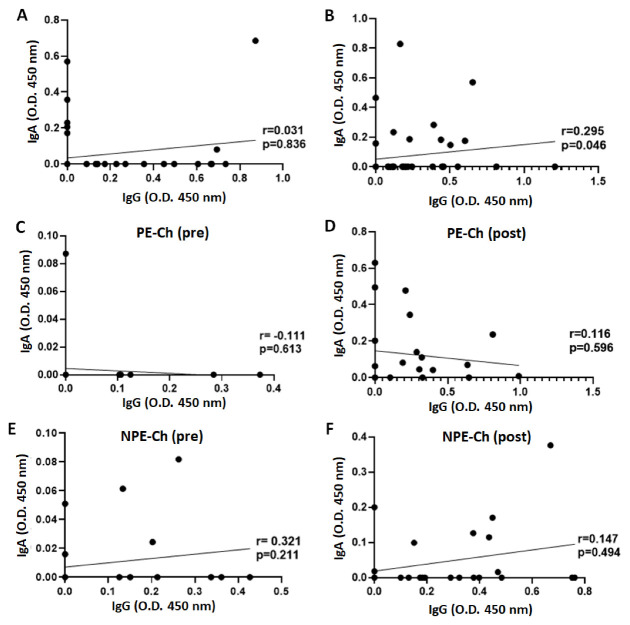
Spearman’s rank correlation coefficient analysis between serum levels of IgA and IgG in the volunteer groups before (pre, **A**,**C**,**E**) and after 30 days of the administration of the second dose (post, **B**,**D**,**F**) of CoronaVac (PE-Co, **A**,**B**) or ChadOx-1 (PE-Ch—**C**,**D**; NPE-Ch—**E**,**F**) vaccines.

**Figure 7 ijerph-20-01939-f007:**
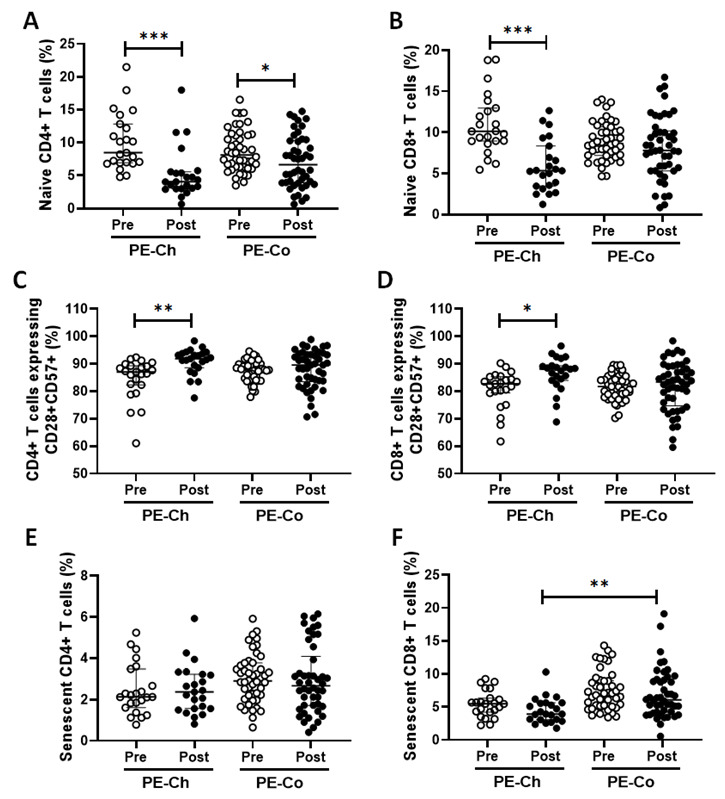
Percentages of naive (CD28+CD57-), double-positive (CD28+CD57+), and senescent (CD28-CD57+) cells, both for CD4+ T cells (**A**,**C**,**E**, respectively) and CD8+ T cells (**B**,**D**,**F**, respectively) in the older adults of the PE group, both before (pre) and 30 days after (post) the administration of the second dose of vaccines ChadOx-1 (PE-Ch) or CoronaVac (PE-Co). * *p* < 0.5; ** *p* < 0.01; *** *p* < 0.001.

**Table 1 ijerph-20-01939-t001:** Characterization of participants by the three intervention groups for biosocial and anthropometric data.

	Physical Exercise Group (PE)		Non-Physical Exercise Group (NPE)	
	CoronaVac Vaccine (*n* = 46)	ChadOx-1 Vaccine (*n* = 23)	ChadOx-1 Vaccine (*n* = 24)	*p*-Value
Age (years)	74.4 ± 3.9	75.3 ± 9.1	75.6 ± 7.9	>0.05
Women (n)	34	18	06	<0.0001
Men (n)	12	05	18	<0.0265
Height (m)	1.57 ± 0.1	1.57 ± 0.08	1.56 ± 0.09	>0.05
Weight (kg)	63.4 ± 12.2	65.9 ± 12.7	66.7 ± 16.6	>0.05
Body mass index	25.9 ± 4.6	26.7 ± 5.3	27.4 ± 6.2	>0.05

Notes: data presented as mean and standard deviation (M ± SD), as well as the number of women and men who composed intervention groups; n = sample; m = meters; kg = kilograms; Body mass index = kg/m^2^ formula.

## Data Availability

Data availability is under the responsibility of the authors and any data can be made available upon request and need.

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
