# Peer review of "Older Adults Who Maintained a Regular Physical Exercise Routine before the Pandemic Show Better Immune Response to Vaccination for COVID-19"

_ijerph, 2023, doi:10.3390/ijerph20031939_

Round 1

Reviewer 1 Report

1.      Introduction

Line 53 - remove "when".

Line 55 - replace side effects with symptoms.

Line 58 - In fact, one of the biggest contributors to the worsening of COVID-19 is associated with an exacerbated immune response and not, exactly, a decrease in immune activity. Although immunosenescence can contribute to the worsening of many infections, and even COVID-19, inflammatory processes seem to contribute more to severe COVID-19. In this case, inflammaging, a process where a low degree of chronic inflammation occurs in elderly people, seems to be a good contributor to this aggravation. However, as the article deals for vaccination, immunosenescence can negatively contribute to the formation of a protective immune response. Therefore, I think it is worth making this distinction in the introduction.

2.      Material and methods

Line 119 - 2.1. Study desing - From the description of the project, it is assumed that it would be destined to the study of another type of vaccine, since, 18 months before the pandemic, when the exercise intervention was started, there was no idea that a pandemic would start in March 2020. I think it would be interesting to clarify this situation, because, initially, I believed that the exercise, to which the work refers, would not have been controlled by you.

“Despite the limitations imposed by the pandemic period, the all three experimental groups were encouraged to maintain their daily activity routines normally”.  - I think this is a limitation of the work, because not all people will have performed the same volume and intensity of exercises, since during this period, the exercised was not controlled.

Line 197 – 2.6. Blood sample collection - Did you only evaluate IgA and IgG antibody levels against the SARS-CoV-2 antigens in the first blood sample? Was the immunophentyping only performed on the second blood sample?

This was not made clear in the text. Please, clarify this.

Line 198 – Blood sample collected before (pre) was collected in January-February 2020 (before the COVID-19 pandemic began)?

Although the authors recognize the limitations of the work, with which we agree, it is still possible to point out others: (1) The time that the individuals remained without physical exercise (at least 1 year) before vaccination. As the effects of physical exercise depend on its continuous practice, 12 months without this practice can really impact the previously achieved effects. In addition, as the volunteers were encouraged to remain physically active during isolation, there is no way of knowing how many followed this recommendation and how many did not, which could have an impact on the results obtained. (2) The impossibility of guaranteeing that individuals have not had previous infections with SARS-CoV2, or even with other coronaviruses that cross-react with SARS-CoV-2 antigens. As some individuals have detectable levels of antibodies even before vaccination, this possibility seems to be present. (3) The lack of a sedentary control group that had been vaccinated with CoronaVac. It makes no sense for a group exercised and vaccinated with CoronaVac if we cannot compare it with sedentary individuals vaccinated with the same vaccine, since the work aims to verify the effect of physical exercise on vaccination against COVID-19. (4) The lack of T lymphocyte immunophenotyping in the NPE-Ch group (and also in an NPE-Co group, non-existent) prevents verification of the effect of physical exercise on the cellular immune response elicited by vaccination.

We also reinforce the significant difference between men and women, as women respond better to vaccination. A greater number of men are found in the sedentary group, which may compromise the result, precisely because this group showed worse performance in the vaccine response. How to make sure that the effect is only due to the practice of physical exercise performed 1 year before vaccination and is not due to most of the volunteers are female in the PE-Ch group, presenting a better vaccinial response, while in the NPE-Ch group most are men?

Although the central idea of the work is very interesting and important, the conditions for carrying it out were greatly impaired, as the pandemic broke the normal progress that one would expect from such work. In this case, the exercise practice period was very distant from the time of vaccination and important controls were not performed (NPE-Co and immunophenotyping of T lymphocytes in the NPE-Ch and NPE-Co groups, the latter of which did not even exist). Unfortunately, these controls are impossible to realize now.

Author Response

Comments and Suggestions for Authors

- Authors´ comments: First of all, we sincerely would like to thank you for the criticism and also for your constructive comments/suggestions that enable us to improve this study. In addition, it is noteworthy to clarify that all alterations concerning your comments/suggestions in the main text are marked in red.

  1. Introduction

Line 53 - remove "when".

- Authors´ response: We would like to thank you for the recommendation and to inform you that the word "when" was removed from the text.

Line 55 - replace side effects with symptoms.

- Authors´ response: We would like to thank you for the suggestion and, as recommended, we replace the side effects with symptoms.

Line 58 - In fact, one of the biggest contributors to the worsening of COVID-19 is associated with an exacerbated immune response and not, exactly, a decrease in immune activity. Although immunosenescence can contribute to the worsening of many infections, and even COVID-19, inflammatory processes seem to contribute more to severe COVID-19. In this case, inflammaging, a process where a low degree of chronic inflammation occurs in elderly people, seems to be a good contributor to this aggravation. However, as the article deals for vaccination, immunosenescence can negatively contribute to the formation of a protective immune response. Therefore, I think it is worth making this distinction in the introduction.

- Authors´ response: We would like to thank you for the comment and to inform you that, as recommended, we revised the two first sentences of the "Introduction" section in order to provide a better distinction of the importance of both phenomena immunosenescence and inflammaging in the context of COVID-19 and older adult population, as presented below.

"...However, it is known that the severity of COVID-19 can be closely associated not only with some clinical conditions of the individual infected by the SARS-CoV-2 virus but also with an exacerbated immune/inflammatory reaction to this infection, which induces a systemic hyperinflammatory response known as the “Cytokine Storm” [5]. Based on the fact that inflammatory response can be involved in the severity of SARS-CoV-2 infection, epidemiological data highlight that older adults are among the populations most affected by COVID-19, in part, due to the presence of a phenomenon named "inflammaging", which is characterized as a chronic, sterile low-grade systemic inflammatory profile and that can favour this population to present the highest rates of severe disease and death from SARS-CoV-2 infection [6].

In agreement with the literature, the occurrence of another phenomenon named immunosenescence, which is characterized by the gradual and significant reduction in immune responses associated with aging, is a pivotal factor that can also lead this population to present high fatality rates COVID-19-related [7, 8]..."

  1. Material and methods

Line 119 - 2.1. Study desing - From the description of the project, it is assumed that it would be destined to the study of another type of vaccine, since, 18 months before the pandemic, when the exercise intervention was started, there was no idea that a pandemic would start in March 2020. I think it would be interesting to clarify this situation, because, initially, I believed that the exercise, to which the work refers, would not have been controlled by you.

- Authors´ response: We would like to thank you for the comment and to inform you that, as recommended, we revised the subsection "2.1. Study design" in order to clarify it, as presented below.

"Data and blood samples were collected between January-February 2021 (before the administration of the first dose of vaccine for COVID-19) and between March-April 2021 (30 days after the administration of the second dose of CoronaVac vaccine) or between May-June 2021 (30 days after the administration of the second dose of ChadOx-1 vac-cine). It is noteworthy to mention that this difference in terms of sample collection was associated with the vaccination schedule for CoronaVac and ChadOx-1 since for CoronaVac the second dose was administrated around 28 days after the first dose was received, and for ChadOx-1 the second dose was administrated around 120 days after the first dose was received."

“Despite the limitations imposed by the pandemic period, the all three experimental groups were encouraged to maintain their daily activity routines normally”.  - I think this is a limitation of the work, because not all people will have performed the same volume and intensity of exercises, since during this period, the exercised was not controlled.

- Authors´ response: We would like to thank you for the comment and to inform you that this sentence was added with the intention to clarify that we did not orient or recommended any alteration in the daily habit of the volunteers enrolled in this study during the period of blood sampling collection since any alteration could impact in the results obtained here. In addition, it is paramount to mention that during the study period, all volunteers did not perform exercise training. Therefore, the control of exercise training was not necessary. However, we would like to inform you that we revised this sentence and moved it to the subsection "2.3. Participants' selection criteria", as presented below (:

"In addition, it is also important to cite that, during the study, period all volunteers were oriented to maintain their daily activity routines."

Line 197 – 2.6. Blood sample collection - Did you only evaluate IgA and IgG antibody levels against the SARS-CoV-2 antigens in the first blood sample? Was the immunophentyping only performed on the second blood sample?

This was not made clear in the text. Please, clarify this.

- Authors´ response: We would like to thank you for the comment and to inform you that, as recommended, we revised the subsection "2.6. Blood sample collection" in order to clarify it, as presented below:

“Blood samples were collected on two different occasions: before (pre) and 30 days after (post) administration of the second dose of the vaccine for COVID-19, both for CoronaVac and for ChadOx-1. One blood sample was collected in a tube without any anticoagulant compound to obtain sera aliquots, which were used to determine the specific IgA and IgG antibodies levels for SARS-CoV-2 antigens. Two other blood samples were collected in tubes containing anticoagulant EDTA to obtain the peripheral blood mononuclear cells, which were used in T cells immunophenotyping analysis, as followed described.”

Line 198 – Blood sample collected before (pre) was collected in January-February 2020 (before the COVID-19 pandemic began)?

- Authors´ response: We would like to thank you for the comment and to apologize for this incorrect information. We revised this sentence, as presented below:

"Data and blood samples were collected between January-February 2021 (before the administration of the first dose of vaccine for COVID-19) and between March-April 2021 (30 days after the administration of the second dose of CoronaVac vaccine) or between May-June 2021 (30 days after the administration of the second dose of ChadOx-1 vaccine)."

Although the authors recognize the limitations of the work, with which we agree, it is still possible to point out others: (1) The time that the individuals remained without physical exercise (at least 1 year) before vaccination. As the effects of physical exercise depend on its continuous practice, 12 months without this practice can really impact the previously achieved effects. In addition, as the volunteers were encouraged to remain physically active during isolation, there is no way of knowing how many followed this recommendation and how many did not, which could have an impact on the results obtained.

- Authors' response: We would like to thank you for the comment and to clarify that one of the main purposes of the study was really to verify whether the regular practice of exercise training by long-standing, which, as formerly reported by our group, is able to improve the immune response to vaccination in this population, could favor the COVID-19 vaccination or not. Although our results were interesting, we never affirm that these data were derived from exercise training, we only suggested that the significant differences obtained in the PE group could be favored by the long-standing exercise training. In addition, we corrected the information that the volunteers were encouraged to remain physically active during isolation, we only recommended to maintain their daily habits during the study period.

(2) The impossibility of guaranteeing that individuals have not had previous infections with SARS-CoV2, or even with other coronaviruses that cross-react with SARS-CoV-2 antigens. As some individuals have detectable levels of antibodies even before vaccination, this possibility seems to be present.

- Authors' response: We would like to thank you for the comment and, as suggested, a new limitation of the study associated with this issue was added in the main text, as presented below:

"3) the lack of information on whether the participants who presented specific levels of IgG and IgA for SARS-CoV-2 antigens before the COVID-19 vaccination were really infected by SARS-CoV-2 or other coronaviruses that cross-react with SARS-CoV-2 antigens."

(3) The lack of a sedentary control group that had been vaccinated with CoronaVac. It makes no sense for a group exercised and vaccinated with CoronaVac if we cannot compare it with sedentary individuals vaccinated with the same vaccine, since the work aims to verify the effect of physical exercise on vaccination against COVID-19.

- Authors' response: We would like to thank you for the suggestion and a new limitation of the study regarding to this issue was added in the main text, as presented below:

"2) a group of older adults’ non-practitioners of exercise training vaccinated with CoronaVac, which could not only improve our understanding of the immunogenicity of this vaccine in the older adult population but also verify whether the regular practice of exercise training by long-standing could help the older adults to present a satisfactory immune response to the vaccination with CoronaVac."

(4) The lack of T lymphocyte immunophenotyping in the NPE-Ch group (and also in an NPE-Co group, non-existent) prevents verification of the effect of physical exercise on the cellular immune response elicited by vaccination.

- Authors' response: We would like to thank you for the comment and to clarify that, specifically in relation to the NPE group, due to the fact that these volunteers living in the hospital and also that during the study period the access to some locals were prohibited, we had no total access to the blood samples of them. In fact, it was provided only one blood sample collected in a tube without an anticoagulant, which allowed us to obtain serum, to perform the present study. Based on it, we cannot perform this analysis (T lymphocyte immunophenotyping) in the NPE. However, in agreement with your comment, a new limitation of the study regarding this issue was added in the main text, as presented below:

"4) the lack of comparison of T lymphocyte immunophenotyping between PE and NPE groups, which could improve the present study findings."

We also reinforce the significant difference between men and women, as women respond better to vaccination. A greater number of men are found in the sedentary group, which may compromise the result, precisely because this group showed worse performance in the vaccine response. How to make sure that the effect is only due to the practice of physical exercise performed 1 year before vaccination and is not due to most of the volunteers are female in the PE-Ch group, presenting a better vaccinial response, while in the NPE-Ch group most are men?

- Authors' response: We would like to thank you for the comment and to inform you that, in order to clarify this issue, we performed new analyses associated with the specific antibody (both IgG and IgA) response for SARS-CoV-2 antigens in the volunteer groups submitted to vaccination with ChadOx-1 separating they by gender. The data obtained were presented in a supplementary figure and it was possible to observe that gender did not impact the vaccination response since no differences between the older men and older women in both volunteer groups (NPE and PE) were found. The significant differences were found exclusively in the intragroup analysis.

Although the central idea of the work is very interesting and important, the conditions for carrying it out were greatly impaired, as the pandemic broke the normal progress that one would expect from such work. In this case, the exercise practice period was very distant from the time of vaccination and important controls were not performed (NPE-Co and immunophenotyping of T lymphocytes in the NPE-Ch and NPE-Co groups, the latter of which did not even exist). Unfortunately, these controls are impossible to realize now.

- Authors' response: We would like to thank you for the comment and, based on the formerly responses to the questions raised by the reviewer, we added not only new information but also a new figure to the manuscript, everything to improve our study. Based on it, we hope that the manuscript can be accepted for publication since it can drive others future research to be performed.

Reviewer 2 Report

The proposed manuscript is a well-prepared, comprehensive and concise review, that aims to analyze the positive effects of physical exercise on the immune response after vaccination for COVID-19 in an aged population.

The introduction reveals what is already known about this topic and the authors clearly explain the starting background, including appropriate studies.

Successive sections are explained in a very detailed and comprehensive way, providing the reader with adequate information.

Figures and tables correlate with and support the discussion.

I have only one suggestion: the authors should elaborate on the physical exercise aspect, perhaps even citing works that have addressed this type, such as:

1.     Brancaccio M, Mennitti C, Gentile A, Correale L, Buzzachera CF, Ferraris C, Montomoli C, Frisso G, Borrelli P, Scudiero O. Effects of the COVID-19 Pandemic on Job Activity, Dietary Behaviours and Physical Activity Habits of University Population of Naples, Federico II-Italy. Int J Environ Res Public Health. 2021 Feb 5;18(4):1502. doi: 10.3390/ijerph18041502. PMID: 33562476; PMCID: PMC7915794.

2.     Nieman DC. Exercise Is Medicine for Immune Function: Implication for COVID-19. Curr Sports Med Rep. 2021 Aug 1;20(8):395-401. doi: 10.1249/JSR.0000000000000867. PMID: 34357885.

3.     Scudiero O, Lombardo B, Brancaccio M, Mennitti C, Cesaro A, Fimiani F, Gentile L, Moscarella E, Amodio F, Ranieri A, Gragnano F, Laneri S, Mazzaccara C, Di Micco P, Caiazza M, D'Alicandro G, Limongelli G, Calabrò P, Pero R, Frisso G. Exercise, Immune System, Nutrition, Respiratory and Cardiovascular Diseases during COVID-19: A Complex Combination. Int J Environ Res Public Health. 2021 Jan 21;18(3):904. doi: 10.3390/ijerph18030904. PMID: 33494244; PMCID: PMC7908487.

4.     Xu Z, Chen Y, Yu D, Mao D, Wang T, Feng D, Li T, Yan S, Yu Y. The effects of exercise on COVID-19 therapeutics: A protocol for systematic review and meta-analysis. Medicine (Baltimore). 2020 Sep 18;99(38):e22345. doi: 10.1097/MD.0000000000022345. PMID: 32957405; PMCID: PMC7505377.

Author Response

Comments and Suggestions for Authors

The proposed manuscript is a well-prepared, comprehensive and concise review, that aims to analyze the positive effects of physical exercise on the immune response after vaccination for COVID-19 in an aged population.

The introduction reveals what is already known about this topic and the authors clearly explain the starting background, including appropriate studies.

Successive sections are explained in a very detailed and comprehensive way, providing the reader with adequate information.

Figures and tables correlate with and support the discussion.

- Authors´ comments: We sincerely would like to thank you for the criticism and also for your constructive comments/suggestions that enable us to improve this study. In addition, it is noteworthy to clarify that all alterations concerning your comments/suggestions in the main text are marked in blue.

I have only one suggestion: the authors should elaborate on the physical exercise aspect, perhaps even citing works that have addressed this type, such as:

  1. Brancaccio M, Mennitti C, Gentile A, Correale L, Buzzachera CF, Ferraris C, Montomoli C, Frisso G, Borrelli P, Scudiero O. Effects of the COVID-19 Pandemic on Job Activity, Dietary Behaviours and Physical Activity Habits of University Population of Naples, Federico II-Italy. Int J Environ Res Public Health. 2021 Feb 5;18(4):1502. doi: 10.3390/ijerph18041502. PMID: 33562476; PMCID: PMC7915794.
  2. Nieman DC. Exercise Is Medicine for Immune Function: Implication for COVID-19. Curr Sports Med Rep. 2021 Aug 1;20(8):395-401. doi: 10.1249/JSR.0000000000000867. PMID: 34357885.
  3. Scudiero O, Lombardo B, Brancaccio M, Mennitti C, Cesaro A, Fimiani F, Gentile L, Moscarella E, Amodio F, Ranieri A, Gragnano F, Laneri S, Mazzaccara C, Di Micco P, Caiazza M, D'Alicandro G, Limongelli G, Calabrò P, Pero R, Frisso G. Exercise, Immune System, Nutrition, Respiratory and Cardiovascular Diseases during COVID-19: A Complex Combination. Int J Environ Res Public Health. 2021 Jan 21;18(3):904. doi: 10.3390/ijerph18030904. PMID: 33494244; PMCID: PMC7908487.
  4. Xu Z, Chen Y, Yu D, Mao D, Wang T, Feng D, Li T, Yan S, Yu Y. The effects of exercise on COVID-19 therapeutics: A protocol for systematic review and meta-analysis. Medicine (Baltimore). 2020 Sep 18;99(38):e22345. doi: 10.1097/MD.0000000000022345. PMID: 32957405; PMCID: PMC7505377.

      - Authors´ comments: We would like to thank you for the suggestion and to inform you, as recommended, we revised the main text and new sentences concerning the issues presented in these references were added in the "Introduction" and "Discussion" sections and are highlighted in blue color.

Reviewer 3 Report

The work is of great scientific relevance and has merit for publication. English revision only required.

Author Response

Comments and Suggestions for Authors

The work is of great scientific relevance and has merit for publication. English revision only required.

- Authors´ comments: We would like to thank you for the comment and to inform you that, as recommended, an English revision was performed.